# Topological defects in a double-mirror quadrupole insulator displace diverging charge

**Isidora Araya Day**[1*], **Anton R. Akhmerov**[2] **and Dániel Varjas**[3]

**1** QuTech and Kavli Institute of Nanoscience, Delft University of Technology,
Delft 2600 GA, The Netherlands
**2** Kavli Institute of Nanoscience, Delft University of Technology,
P.O. Box 4056, 2600 GA Delft, The Netherlands
**3** Department of Physics, Stockholm University, AlbaNova University Center,
106 91 Stockholm, Sweden

⋆ i.araya.day@gmail.com

## Abstract

We show that topological defects in quadrupole insulators do not host quantized fractional charges, contrary to what their Wannier representation indicates. In particular, we test the charge quantization hypothesis based on the Wannier representation of a disclination and a parametric defect. Since disclinations necessarily strain the lattice and parametric defects require closed curves in parameter space, both defects break four-fold rotation symmetry, even away from their origin. The Wannier representation of the defects is thus determined by local reflection symmetries. Contrary to the hypothesis, we find that the local charge density decays as $\sim 1/r^2$ with distance, leading to a diverging defect charge. Because topological defects are incompatible with four-fold rotation symmetry, we conclude that defect charge quantization is protected by sublattice symmetry, and not higher order topology.

**See also:** online presentation recording.

A topological quadrupole insulator hosts quantized half-integer corner charges [1–5], with the Benalcazar-Bernevig-Hughes (BBH) model being a canonical example of this phase (see Fig. 1(a)). In the bulk a quantized quadrupole moment serves as a topological invariant, providing the quadrupole insulator with a bulk-corner correspondence [6, 7]. While the precise definition of the bulk quadrupole moment is a subtle issue [8, 9], the bulk-corner correspondence is guaranteed by the combination of the four-fold rotation symmetry and two anti-commuting reflection symmetries.

Disclinations in quadrupole insulators were proposed to serve as bulk probes of topology, emulating the behavior of the corners by trapping quantized fractional charges [10]. Furthermore, Refs. [10, 11] demonstrated that if a band insulator admits a real space Wannier representation—*i. e.* has a vanishing Chern number—then the non-uniform distribution of the Wannier centers establishes the corner and defect charges quantization. The topology of a

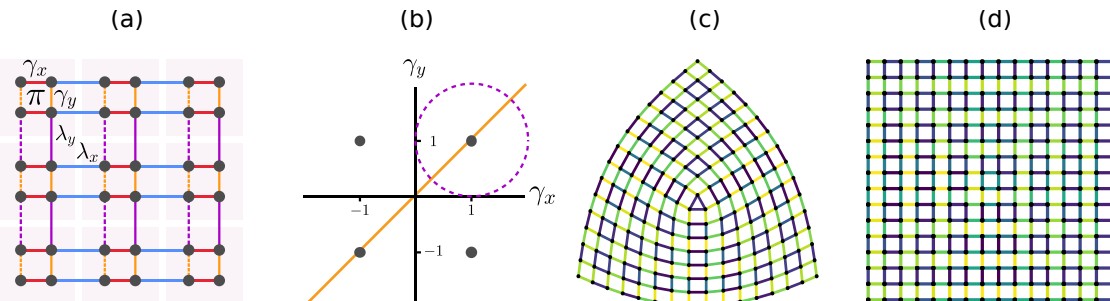

Figure 1: (a) The BBH model has four orbitals per unit cell, connected via $\gamma_{x,y}$ and $\lambda_{x,y}$ hoppings. The dashed hopping lines have an extra phase of $-1$ with respect to the solid lines, threading a $\pi$ flux through each plaquette. (b) The phase diagram shows that the bulk gap closes at the grey dots, when $|\gamma_x/\lambda_x| = 1$ and $|\gamma_y/\lambda_y| = 1$. The orange line indicates the set of parameters that preserve four-fold rotation symmetry and the dashed purple curve is a circular gapped path centered in $(\gamma_x, \gamma_y) = (1,1)$ with radius $\gamma_r = 1$. (c) A disclination is created by effectively removing one orbital from a unit cell. We expect both defects to host localized fractional charges. (d) A parametric defect is inserted in the BBH model using a gapped path in the phase diagram. The parametrization of the hoppings shifts the strongly bounded unit cells around the defect, creating a charge deficiency at it. The hopping strengths are given by the loop shown in (b).

quadrupole insulator was recently probed via disclination defects, successfully finding defect bound states in microwave metamaterials [12].

An alternative pathway to creating topological defects, so far not explored in the context of quadrupole insulators, is to make the Hamiltonian parameters position-dependent [13,14] by making the Hamiltonian $H(\boldsymbol{k}, \boldsymbol{r})$ gradually vary with position. In conventional topological phases, as long as $H(\boldsymbol{k}, \boldsymbol{r})$ remains gapped far away from the defect, the presence of a defect bound state is protected by the defect invariant. Breaking the four-fold rotation symmetry while preserving both reflection symmetries allows us to introduce non-contractible loops in the space of the Hamiltonians of a quadrupole insulator [15], see Fig. 1(b). Therefore, similarly to the quantized disclination charge, one may expect that the charge of a topological parametric defect is quantized in a quadrupole insulator.

The topological arguments presented in earlier works, as well as the Wannier center considerations above suggest that the defect charge quantization is a direct consequence of topology. There is, however, no rigorous proof that the defect charge quantization is indeed a consequence of the quantized quadrupole moment. Our goal, therefore, is to put the charge quantization hypothesis to a test and rigorously identify the conditions under which the defect charge is quantized.

To test the hypothesis we use the BBH tight-binding model: a two-dimensional lattice with four orbitals per unit cell. Figure 1(a) shows the model, which corresponds to a quadrupole insulator that admits a Wannier representation, and whose Bloch Hamiltonian is:

$$\mathcal{H}(\boldsymbol{k}) = [\gamma_x + \lambda_x \cos(k_x)]\Gamma_4 + \lambda_x \sin(k_x)\Gamma_3 \\ + [\gamma_y + \lambda_y \cos(k_y)]\Gamma_2 + \lambda_y \sin(k_y)\Gamma_1, \tag{1}$$

where $\gamma_{x,y}$ and $\lambda_{x,y}$ account for intra-cell and inter-cell hopping amplitudes, respectively. Here $\Gamma_k = -\sigma_2 \otimes \tau_k$ for $k \in \{1,2,3\}$, and $\Gamma_4 = \sigma_1 \otimes \tau_0$, with $\sigma_i$ and $\tau_i$ Pauli matrices acting on the orbital degrees of freedom.

We introduce a disclination by removing the quarter of the lattice that is spanned by $\theta \in [3\pi/2, 2\pi)$, changing the positions of the remaining sites according to $\theta \to 4/3\theta$, while

keeping $r \equiv |\boldsymbol{r}|$ constant (see Fig. 1(c)). We keep the lattice isotropic by choosing $\gamma_x = \gamma_y = \gamma$ and $\lambda_x = \lambda_y = \lambda$. If the disclination center lies between the strongly coupled sites, the disclination removes an orbital from one of the strongly coupled clusters. The Wannier centers are located at the center of these clusters, such that the resulting strongly bonded triangle has 3/2 electrons at half-filling, and therefore a half-integer charge. An alternative way to come to the same conclusion is by observing that each of the three corners must have a charge 1/2 because the quadrupole insulator is in its topological phase. The remaining half integer charge must then be bound to the disclination center—the only remaining lattice defect.

Both the argument above and the proof of Ref. [10] ignore the effect of lattice distortion around the defect center. To study the effect of strain on the tight-binding model, we modulate the hopping amplitudes by changing them proportionally to the bond length:

$$\gamma'(\boldsymbol{r}_1, \boldsymbol{r}_2) = \gamma(1 + \alpha_\gamma(|\boldsymbol{r}_1 - \boldsymbol{r}_2| - a)),$$
$$\lambda'(\boldsymbol{r}_1, \boldsymbol{r}_2) = \lambda(1 + \alpha_\lambda(|\boldsymbol{r}_1 - \boldsymbol{r}_2| - a)), \tag{2}$$

where $\gamma'$ and $\lambda'$ are hopping amplitudes coupling the orbitals that would be connected by a hopping $\gamma$ or $\lambda$ without a disclination, $a = 1/2$ is the bond rest length, and $\boldsymbol{r}_1$, $\boldsymbol{r}_2$ are the coordinates of the two orbitals. The factors $\alpha_\gamma$ and $\alpha_\lambda$ are the electron-lattice couplings. To keep the system in the topological phase, we require that far away from the disclination the ratio $|\gamma/\lambda| < 1$. Because the disclination necessarily introduces strain, without fine-tuning that ensures $\alpha_\gamma = \alpha_\lambda = 0$, the disclination necessarily breaks the four-fold rotation symmetry even far away from its center.

A parametric defect is an alternative way to create a quantized fractional charge. We create this defect in the BBH model at an arbitrary lattice coordinate $\boldsymbol{r}_0$. We fix $\lambda_x = \lambda_y = 1$ and vary the hopping strengths $\gamma_x$ and $\gamma_y$ with displacement $\boldsymbol{r}$ from the defect. For simplicity we consider a defect where $\gamma_x$ and $\gamma_y$ only depend on the angle $\theta$ between $\boldsymbol{r}$ and the $x$-axis. We require that as the angle $\theta$ of $\boldsymbol{r}$ in polar coordinates advances by $2\pi$, the vector $(\gamma_x, \gamma_y)$ encloses a loop around the point $(1, 1)$. As an example, Fig. 1(d) shows the hopping strengths corresponding to the circular loop in the parameter space shown in purple in Fig. 1(b). The center of the defect has an odd-sized cluster of strongly coupled sites, which contains a half-integer number of electrons at half-filling, similarly to a disclination. Tracking the positions of the Wannier centers within the unit cell [15] as a function of $\boldsymbol{r}$, shown in Fig. 2, demonstrates the presence of the defect charge.

While the defect charge quantization hypothesis relies only on the spatial symmetry arguments, the BBH model Eq. (1) is minimal and therefore it has additional symmetries. In particular, the sublattice symmetry of the BBH model by itself leads to charge quantization. To see this we consider the local charge density in a unit cell

$$\rho_{ij} = \sum_{E_l < E_F} \sum_\alpha |\Psi_{\alpha l}^{ij}|^2, \tag{3}$$

where $E_l \leq E_F \equiv 0$ labels the occupied energies with the Fermi level fixed in the bulk gap and $\alpha \in \{1, 2, 3, 4\}$ labels the four orbitals of the unit cell. The indices $i, j$ are the lattice coordinates of a unit cell, where the defect origin is at $i = 0, j = 0$, and $\Psi_{\alpha l}^{ij} = \langle i, j, \alpha | \Psi_l \rangle$. We express the total number of states in a unit cell through the contributions of eigenstates at different energies:

$$4 = \sum_{E_l < 0} \sum_{\alpha=1}^4 |\Psi_{\alpha l}^{ij}|^2 + \sum_{E_l = 0} \sum_{\alpha=1}^4 |\Psi_{\alpha l}^{ij}|^2 + \sum_{E_l > 0} \sum_{\alpha=1}^4 |\Psi_{\alpha l}^{ij}|^2$$
$$= 2 \sum_{E_l < 0} \sum_{\alpha=1}^4 |\Psi_{\alpha l}^{ij}|^2 + \sum_{E_l = 0} \sum_{\alpha=1}^4 |\Psi_{\alpha l}^{ij}|^2 = 2\rho_{ij} + \sum_{E_l = 0} \sum_{\alpha=1}^4 |\Psi_{\alpha l}^{ij}|^2, \tag{4}$$

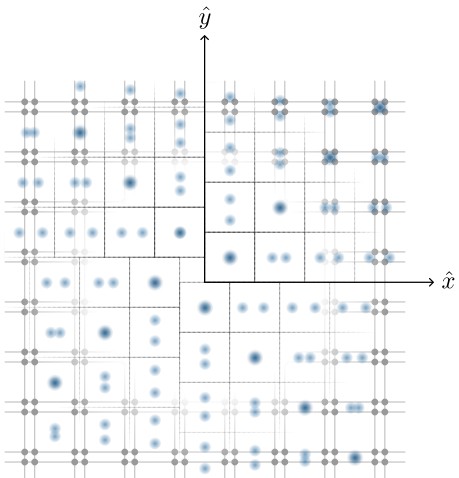

Figure 2: The parametric defect interpolates between the trivial (top right quadrant) and topological (bottom left quadrant) phases of the BBH model (grey dots connected by lines) by smoothly modulating the hoppings and preserving local mirror symmetries. While in the trivial phase the Wannier centers (blue circles) are localized at the center of the unit cell, they are localized at the corners in the topological phase. In all unit cells the Wannier centers come in pairs invariant under local mirror symmetries. The square cells, each centered in between pairs of nearest Wannier centers, tile the space away from the defect. The unoccupied quarter unit cell at $r = 0$ corresponds to the missing fractional charge $e/2$.

where we use the sublattice symmetry in the second equality. Finally, the total defect charge $q_{\text{tot}}$ integrated over a square-shaped area surrounding the defect is the charge excess with respect to the uniform charge density $\rho_{ij} = 2$:

$$q_{\text{tot}}(R) = \sum_{i,j=-R/2}^{R/2} \left( \rho_{ij} - 2 \right). \tag{5}$$

Substituting Eq. (4) into Eq. (5) we obtain

$$q_{\text{tot}}(R) = -\frac{1}{2} \sum_{i,j=-R/2}^{R/2} \sum_{E_l=0} \sum_{\alpha=1}^{4} |\Psi_{\alpha l}^{ij}|^2. \tag{6}$$

Therefore in presence of the sublattice symmetry, the total defect charge is $N_0/2$, where $N_0$ is the number of zero-energy modes localized at the defect. The number of the zero-energy modes, in turn, is defined by the second winding number of the Hamiltonian around the defect, and protected already by the sublattice symmetry alone [14]. Furthermore, because of the bulk gap, the zero-modes are exponentially localized.

Sublattice symmetry also guarantees the disclination charge quantization, despite the center of the disclination locally breaking the sublattice symmetry. To demonstrate this, we use perturbation theory and consider $3/4^{\text{th}}$ of the BBH lattice the unperturbed Hamiltonian. This Hamiltonian inherits sublattice symmetry from the BBH model and therefore it hosts fractional and exponentially localized charges at its six corners. To obtain a disclination, we add a perturbation that consists on the hopping terms that couple orbitals from neighboring unit cells on the position-transformed lattice, such that the lattice is glued. Using that the system is gapped, we conclude that the Fermi level Green's function, and therefore charge density, changes by an

amount that decays exponentially with distance from the disclination's origin. Because sublattice symmetry applies near the sample corners, we conclude there is a 1/2 charge that may only be located near the disclination.

To distinguish the topological properties of the quadrupole insulator from the effect of the sublattice symmetry, we add hoppings with magnitude $\delta$ connecting sites from the same sublattice in the neighboring unit cells. In order to preserve the anti-commuting reflection and the four-fold rotation symmetries of the original model, we choose the value of all these hoppings to be the same. We change the sign of the additional hoppings when gluing the different sides of the wedge at the disclination in order for the cut to be gauge-compatible with the rest of the lattice.

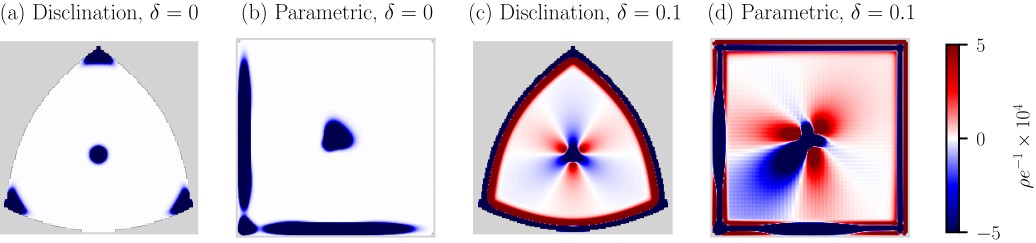

Figure 3: Local charge density at half-filling in the defects. A defect displaces both negative (blue) and positive (red) excess charge in the absence of sublattice symmetry, resulting in a divergent defect charge. (a-b) The parametric defect and the disclination with sublattice symmetry. (c-d) The parametric defect and the disclination without sublattice symmetry.

We numerically study the defect charge inserting the defects at the center of a $L^2 = 50 \times 50$ square system. We create the parametric defect using a circular path of radius $\gamma_r = 1$ centered at $\gamma_x/\lambda_x = \gamma_y/\lambda_y = 1$ (see Fig. 1(b)), and the disclination using $\gamma = \lambda/2 = 0.5$ and $\alpha_\gamma = \alpha_\lambda = \alpha_\delta = 1$. To obtain the defect charges we integrate the charge density distributions shown in Fig. 3(a-d). Our results in Fig. 4(a) confirm that the total defect charge converges to 1/2 in presence of sublattice symmetry, and demonstrate an apparent convergence to a non-quantized value otherwise, unless the electron-lattice coupling is neglected, in agreement with Ref. [10]. This lack of quantization is explained by considering the absolute charge deviation:

$$q(R) = \sum_{i,j=-R/2}^{R/2} \left| \rho_{ij} - 2 \right|. \tag{7}$$

While the absolute charge deviation converges to a finite value in presence of sublattice symmetry, it diverges when the sublattice symmetry and the local four-fold rotation symmetry are broken, as shown in Fig. 4(b). According to our expectations, the defect charge convergence is exponential (Fig. 4(c)). On the other hand, the divergence matches a $\sim 1/r^2$ decay of the local charge density, as confirmed by $\Delta q(R) = q(R+1) - q(R) \propto 1/R$ in Fig. 4(d). The lack of absolute convergence in combination with conditional convergence means that depending on the summation order, the defect charge may assume an arbitrary value according to the Riemann rearrangement theorem. While the absolute convergence of charge deviation is required, it is also a weaker condition on charge quantization than the vanishing charge variance studied in Ref. [16].

In order to confirm the total charge divergence we determine the asymptotic behavior of the local charge density away from the defect. Sufficiently far away from the defect in a large enough sample, the Hamiltonian's position dependence becomes small compared to the

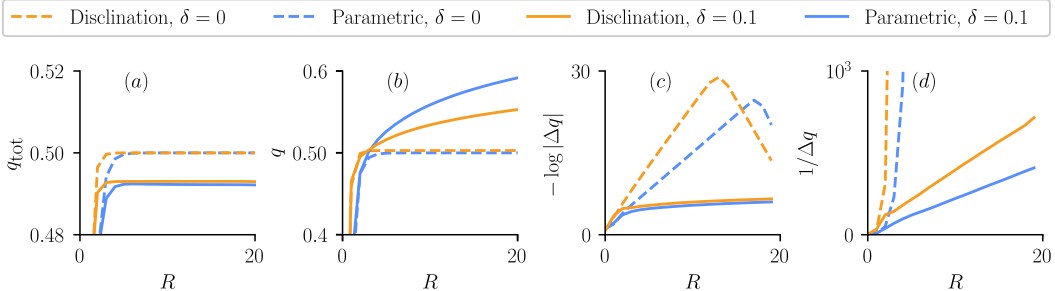

Figure 4: The defect charge diverges in the absence of sublattice symmetry. (a) The total charge $q_{\text{tot}}$ integrates to $1/2$ only when sublattice symmetry is present (dashed lines), otherwise $q_{\text{tot}}$ depends on the integration area (solid lines), demonstrating the lack of absolute convergence. (b) In the presence of sublattice symmetry (dashed lines), the charge deviation $q$ is quantized to $1/2$, otherwise it diverges (solid lines). The divergence of the parametric defect's charge is more pronounced when the paths are asymmetric in parameter space. (c) The defect charge deviation converges exponentially when $\delta = 0$ (dashed lines), as $-\log|q(R+1) - q(R)| \propto R$. (d) For $\delta \neq 0$ the local charge density decays as $1/R^2$ as indicated by $q(R+1) - q(R) \propto 1/R$.

inverse energy gap while it changes slowly in space. Thus, the local real space variations of the Hamiltonian determine the charge density and its local response is captured by perturbation theory that treats the position dependence as a perturbation. To account for $\rho \propto 1/r^2$ found via numerical simulations, it is enough to consider a general position-dependent perturbation expanded to second order around $\boldsymbol{r}^*$ far away from the defect center $\boldsymbol{r}_0$,

$$H'(\delta \boldsymbol{r}) = H'^0 + \delta r_i H_i'^1 + \delta r_i H_{ij}'^2 \delta r_j + h.c., \tag{8}$$

where $H'^0$, $H'^1$ and $H'^2$ are the zeroth, first and second order components of the Hamiltonian perturbation, and $\delta \boldsymbol{r} \equiv \boldsymbol{r} - \boldsymbol{r}^*$. By construction, $H'^0$, $H'^1$, and $H'^2$ are invariant under reflection. While $H'^1$ is reflection-symmetric, $r_i H_i'^1$ is odd under reflection symmetries. The perturbative response of the local charge density is invariant under reflection and it is a power series of $H'^1$ and $H'^2$. As a consequence of being odd under reflection, the first order contribution of $H'^1$ to the charge density vanishes, while the contribution of $H'^2$ and the second order contribution of $H'^1$ remain. Because both types of defects have a finite $dH/d\theta \to \text{const}$ with $r \to \infty$, $H'^1 \sim 1/r$ and $H'^2 \sim 1/r^2$. Therefore, we confirm that the charge density universally decays as $1/r^2$ unless the Hamiltonian is fine-tuned or it has additional symmetries.

In summary, we investigated charge quantization of defect bound states in quadrupole insulators as it was predicted in Refs. [10,11] and reported in Ref. [12]. Specifically, we analyzed a disclination and a parametric defect in the BBH model and found that the previously reported charge quantization is a consequence of the sublattice symmetry, or of neglecting the effects of lattice strain around the disclination. The importance of sublattice symmetry was appreciated in the early studies of soliton-fermion bound states in one-dimensional models [17–19]. This symmetry is not inherent to quadrupole insulators, but without it the defect charge is not quantized, because the absolute charge deviation diverges. Our findings demonstrate that topological protection of the filled bands, despite being characterized by a quantized topological invariant, is weaker than the protection of individual states in topological insulators.

## Acknowledgements

We thank B. Seradjeh for drawing our attention to relevant literature. D. V. is thankful to R. Queiroz for enlightening discussions.

## Data availability

The code used to produce the reported results is available on Zenodo [20].

**Author contributions**   D. V. defined the project goal and formulated the hypothesis. I. A. D. implemented the numerical checks with guidance from D. V. and A. R. A. All authors interpreted the results and contributed to the project planning. I. A. D. and A. R. A. wrote the manuscript with input form D. V.

**Funding information**   This work was supported by the Netherlands Organization for Scientific Research (NWO/OCW), as part of the Frontiers of Nanoscience program and an NWO VIDI grant 016.Vidi.189.180. We acknowledge the support from QuTech Academy through the Scholarship. D. V. was supported by the Swedish Research Council (VR) and the Knut and Alice Wallenberg Foundation.

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
