# Peer review of "Topological defects in a double-mirror quadrupole insulator displace diverging charge"

_SciPost Physics, doi:SciPost Phys. Core 5, 053 (2022)_

## Round 2 · Referee Report · Anonymous (Referee 1) · 2022-5-23

Strengths

This paper provided adequate numerical evidence to demonstrate that the defect charge is not quantized in the double mirror quadrupole insulators. The logical flow is smooth and the argument is easy to follow.

Weaknesses

It didn't directly explain the lack of defect charge quantization through the Wannier representation picture.

Report

The present manuscript studied the quantization of charge in the double-mirror quadrupole insulator (Benalcazar-Bernevig-Hughes model) at the disclination defect. The author provided concrete numerical evidence to demonstrate that without the sublattice symmetry and the four-fold rotation symmetry, the total charge around the defect is not fractionally quantized in units of 1/2. Instead, the local charge density decays as $1/r^2$ when deviating from the defect origin.

The paper is interesting and reveals the importance of sublattice symmetry regarding the quantization of defect charge when additional $C_4$ symmetry is broken. These results are great additions to the previous works [Phys. Rev. B 101, 115115 (2020), Science 368(6495), 1114 (2020)] on bulk and defect correspondence in higher-order topological insulators when $C_n$ symmetry is required. Hence, I would recommend publication.

However, there are some points I would like to encourage the author to consider.

1, For the parametric Hamiltonian, although the bulk remains gapped along the loop, the edge gap closes, and Wannire representation changes. For Fig 2(b), what is the corresponding filling for the charge density? In that case, zero energy states are also localized at the edge and it's unclear if the summation on the right-hand side of eq (7) is quantized.

2, In Fig. 3(a), when sublattice symmetry is broken, the total charge is still converged to a finite value that is close to $1/2$ when $R\rightarrow \infty $. Is there a good physics understanding for that convergence? e.g. is it because the profile of Wannier orbitals loses the symmetry when $C_4$ is broken and portions of Wannier orbitals that fall into the integral region are therefore slightly different from the $1/2$? Would the deviation depend on how large the $C_4$ symmetry is broken?

---

## Round 2 · Referee Report · Anonymous (Referee 2) · 2022-6-14

Strengths

  1. investigation of a fundamental model that exhibits the corner charge
  2. numerical results that the model with defect and without sublattice symmetry does not exhibit quantized defect charge

Weaknesses

  1. absence of discussions on a nontrivial topology indicated by Wannier representation.

Report

In this work, the authors investigated a fundamental model with anticommuting double mirror symmetries, dubbed Benalcazar-Bernevig-Hughes (BBH) model, and discussed defect charge in the presence and absence of sublattice symmetry.

The authors showed that the defect charge is quantized in the presence of chiral symmetry but not in the absence. This implies that the quantization is not robust against symmetry breaking in contrast to ordinally topological phases. This finding matches the first item in expectations of acceptance criteria (https://scipost.org/SciPostPhys/about#criteria).

Although the presenting results are interesting, I would like to clarify the following things.

  1. In Figure 3(a), the converged values q_{tot} of solid lines are almost equal to 0.5. Could the author provide evidence that it is not a numerical error? Or, when the \delta is larger, is the converged value far from 0.5?

  2. In Figures 2 (b-d), it seems that the charge is distributed on the boundary of systems. Do the systems have nontrivial edge states or edge charge?

  3. The authors said "We identify sublattice symmetry and not higher order topology as the origin of the previously reported charge quantization." Then, I wonder if the sublattice symmetry is crucial in the fractional disclination charge of rotation symmetric systems reported in Ref. [10]. Could the authors show the importance of sublattice symmetry through the models in Ref. [10]?

Requested changes

  1. The authors should add the converged value of solid lines in Figure 3(a).
  2. The authors should add plot legends in Figure 2.

  • validity: good
  • significance: good
  • originality: good
  • clarity: good
  • formatting: good
  • grammar: good

Author:  Isidora Araya Day  on 2022-06-15  [id 2583]

(in reply to Report 2 on 2022-06-14)
Category:
question
answer to question

Dear referee,

Thank you for your feedback. We would like to address your inquiry on the applicability of our results to Ref. 10. We would like to note that in addition to having a sublattice symmetry, Ref. 10 disregards the electron-lattice coupling. In other words, despite the dislocations are shown on a curved lattice, the authors consider all hoppings constant, unlike what would happen if one tried to create this Hamiltonian in a real material. Therefore, the charge quantization reported in Ref. 10 is to be expected. If one were to break the sublattice symmetry in the model of Ref 10 and in addition include the electron-lattice couplings, the scaling argument we present in our work (the paragraph containing Eq. 9) would apply in exactly the same way as to the BBH model.

We believe that adding a very similar model (that of Ref. 10) to the manuscript will not improve its readability, and the added value will not justify the extra work. Therefore, we would like to ask if the above explanation addresses your third question sufficiently.

Anonymous on 2022-06-27  [id 2611]

(in reply to Isidora Araya Day on 2022-06-15 [id 2583])
Category:
answer to question

Thank you for your reply.

Actually, I am not still convinced that the sublattice symmetry is essential for the models in Ref. [10]. The reasons are as follows.

  1. The point is whether the rotation symmetry can protect the quantized defect charge. If the sublattice symmetry is essential and the rotation symmetry is not, the rotation symmetry cannot solely quantize the defect charge. Here, although I think that such models have been realized in artificial systems as shown in Ref. [12], it does not matter if the models in Ref. [10] are realistic or not. I am concerned about which symmetries can protect the defect charge.

  2. In the manuscript, the authors emphasized that the sublattice symmetry is the origin of the quantized defect charge, as they said "We identify sublattice symmetry and not higher order topology as the origin of the previously reported charge quantization." In my understanding, "previously reported charge quantization" means the results of Ref. [10]. Since Ref. [10] discussed rotation symmetric systems, the authors should exclude the possibility of the quantization by the rotation symmetry.

  3. Although the authors claimed "We show that topological defects in quadrupole insulators do not host quantized fractional charges, contrary to what their Wannier representation indicates," they did not discuss the Wannier center arguments for the model the authors considered. While the rotation case is obvious, I am not sure that the Wannier center argument is applicable to the double mirror symmetry case.

---

## Round 3 · Referee Report · Anonymous (Referee 2) · 2022-8-12

Report

The authors addressed some of my questions in detail. Also, the authors clarified the quantization of the defect charge in the presence of fourfold rotation.

After reading their reply carefully, I am afraid I feel that the authors oversell their results, and I cannot agree with the authors' statement that C4-symmetry is incompatible with two-dimensional lattices.

More concretely, I have the following concerns.

  1. The authors seem to keep electronic systems in mind and claim that C4-symmetry is incompatible with two-dimensional lattices. However, in my opinion, the discussion in Ref. 10 is also applicable to artificial systems. For example, Ref. 12 discussed metamaterial. Do the authors claim that C4-symmetry should not be realized even in such systems?

  2. Importantly, in metamaterials designed in Ref. 12, the chiral symmetry (equivalently sublattice symmetry) does not exist, as the paper said 
“First, chiral symmetry is broken in the fabricated metamaterials because the capacitive coupling elements effectively increase the electrical length of the resonators, thereby decreasing their resonance frequencies.”
 Nonetheless, the paper reported the quantized defect charge.

In summary, a new finding in this work is to numerically discover the quantized charge solely by sublattice symmetry in a particular model.

On the other hand, I think that the current presentation does not meet the first item in General acceptance criteria. The authors consider only a particular model. Also, the differences/contradiction between this work and references are not adequately clarified. Thus, I am still not convinced that sublattice symmetry is the “unique” origin of the quantized defect charge, as the authors claim.

So, I cannot recommend publication of the manuscript in the present form.

  • validity: good
  • significance: ok
  • originality: good
  • clarity: -
  • formatting: good
  • grammar: good

Author:  Isidora Araya Day  on 2022-12-06  [id 3107]

(in reply to Report 1 on 2022-08-12)
Category:
answer to question

The manuscript was accepted for publication without another review round. However because the referee report lists several issues that question our findings, we would like to respond to the queries for public record.

​1. The authors seem to keep electronic systems in mind and claim that C4-symmetry is incompatible with two-dimensional lattices. However, in my opinion, the discussion in Ref. 10 is also applicable to artificial systems. For example, Ref. 12 discussed metamaterial. Do the authors claim that C4-symmetry should not be realized even in such systems?

Unless a system is fine-tuned, dislocations necessarily introduce slowly decaying inhomogeneities that break spatial symmetries. This consideration applies to metamaterials as well as electronic systems. Artificially designed tight-binding systems where the magnitude of every hopping is controlled individually can be tuned to avoid this.

​2. Importantly, in metamaterials designed in Ref. 12, the chiral symmetry (equivalently sublattice symmetry) does not exist, as the paper said “First, chiral symmetry is broken in the fabricated metamaterials because the capacitive coupling elements effectively increase the electrical length of the resonators, thereby decreasing their resonance frequencies.” Nonetheless, the paper reported the quantized defect charge.

Ref. 12 did not present a quantitative study of how the deviation of the defect density scales with the distance from the origin. Our analysis demonstrates that the subleading terms lead to a logarithmic divergence of the defect charge density that was overlooked in Ref. 12.

In summary, a new finding in this work is to numerically discover the quantized charge solely by sublattice symmetry in a particular model.

We disagree with the refere's summary because it ignores important parts of our work. While the numerical demonstration we presented applies to a specific model, the symmetry analysis and the scaling arguments are universal and apply to all models in this symmetry class, including Ref. 12.

---

## Round 3 · Referee Report · Anonymous (Referee 3) · 2022-9-23

Strengths

  1. The paper gives numerical evidence for a new result, namely that dislocations in quadrupole insulators do not bind a quantized charge.

Weaknesses

  1. The explanation for why the charge is not quantized in the presence of lattice distortion is lacking.

Report

The authors present compelling numerical evidence that a disinclination in a quadrupole insulator need not bind a quantized charge, contrary to earlier work. The authors further show that there is no contradiction since earlier models have sublattice symmetry, while the present work introduces a lattice distortion that breaks sublattice symmetry. The main point is nicely made. However, I do not consider this a "groundbreaking" discovery, per the acceptance criteria of SciPost Physics. I also find the theoretical explanation for the lack of quantization lacking. Thus, I think the paper would be more suitable for SciPost Physics Core, and could be accepted as is: the paper is clearly written and unambiguously demonstrates its main point.

I have a few questions about the lack of quantization: 1. One point that is unclear to me is whether the lattice distortion included by the authors in Eqs (2) and (3) makes the defect long range. In other words, I can't tell from Eqs (2) and (3) if the rate at which the lattice distortion disappears scales much worse than for, e.g., a screw dislocation as considered by Teo and Kane in Ref 14. Could that explain the lack of quantization? 2. The roles of the various symmetries is unclear, leaving me with the following questions: (a) would a lattice distortion that preserved sublattice symmetry have quantized charge? (b) if there was no lattice distortion but sublattice symmetry was broken before the disclination was introduced, would the disinclination have quantized charge? (c) does C4 symmetry breaking play any role in the lack of quantized charge? (Are mirror symmetries enough to provide quantization according to the formulas by Li et al in Ref 10?)

I also have the following minor comments: 1. Fig 1 caption: the caption is worded in a confusing way, where the sentence "We expect both defects..." follows (c), but I think it refers to (c) and (d). I think the caption would be more clear if this sentence was moved to follow part (d). 2. Is Eq (7) the definition of N0? If so, I don't see from the equation why N0 would be an integer. On the other hand, if N0 is defined from Ref 14, then why would it be quantized in some region defined by the length scale R? I might expect it to exponentially decay with R. 3. I don't understand the perturbative argument (spanning p4-5) explaining the quantization provided by sublattice symmetry: is the perturbation gluing the lattice together across the quadrant that was removed? If so, does the gluing consists of "normal" strength bonds, or small strength -- if the bonds are full strength, then is this really a small perturbation? 4. The text sometimes refers to a "divergent defect charge" (e.g., caption Fig 3) and in other places refers to "convergence to a non-quantized value" (e.g., above Eq (8)). Maybe the authors are referring in some places to q and other places q_{tot}. If so, it should be clarified which quantity is being referred to. 5. The authors reference the Riemann rearrangement theorem (top of p6) to argue that the defect charge may assume an arbitrary value. But I am not convinced of this argument: from what I can see, the Riemann rearrangement theorem says that the sum for q_{tot} in Eq (7) can be rearranged to reach an arbitrary value if it is arrange in any order. But I physically, the unit cells closest to the defect should be included in the sum before terms further away. With this constraint, can the sum still reach an arbitrary value? 6. I am confused by the caption to Fig 4(a), which is said to demonstrate the "lack of absolute convergence" for q_{tot}. But Fig 4(a) looks like it converges to a flat line. Perhaps the authors mean a lack of quantization?

  • validity: high
  • significance: good
  • originality: high
  • clarity: good
  • formatting: excellent
  • grammar: good

Author:  Isidora Araya Day  on 2022-12-07  [id 3108]

(in reply to Report 2 on 2022-09-23)

The manuscript was accepted for publication without another review round. However because the referee report lists several issues that question our findings, we would like to respond to the queries for public record.

​1. One point that is unclear to me is whether the lattice distortion included by the authors in Eqs (2) and (3) makes the defect long range. In other words, I can't tell from Eqs (2) and (3) if the rate at which the lattice distortion disappears scales much worse than for, e.g., a screw dislocation as considered by Teo and Kane in Ref 14. Could that explain the lack of quantization?

Our results are consistent with those of Teo and Kane. Reference 14 only considers 0-dimensional defects in presence of particle-hole symmetry or sublattice symmetry. Either of the two protects the zero mode, and guarantees quantization.

Slowly decaying strain caused by a screw dislocation does not remove symmetry-protected zero modes that were a focus of Ref. 14.

​2. (a) Would a lattice distortion that preserved sublattice symmetry have quantized charge?

Yes, lattice distortion that preserved sublattice symmetry would be enough to quantize the defect charge. This is shown by Equations (5-7) and we have confirmed it numerically.

​2. (b) If there was no lattice distortion but sublattice symmetry was broken before the disclination was introduced, would the disinclination have quantized charge?

If there was no lattice distortion, there would be local four-fold rotation symmetry everywhere in the lattice away from the defect origin. As a consequence, the defect charge would be quantized independently of the presence of sublattice symmetry, as proved and demonstrated in the previous works. We have also confirmed numerically that this is the case. However, disclinations in purely 2D systems necessarily introduce strain, and therefore break four-fold rotation. Taking strain into account is what leads to a divergent charge.

​2. (c) Does C4 symmetry breaking play any role in the lack of quantized charge?

Yes, a local four-fold rotation symmetry quantizes the defect charge. However, it is incompatible with two-dimensional disclinations. Breaking four-fold rotation symmetry by strain leaves the defect charge quantization up to the presence of sublattice symmetry.

​2. (d) Are mirror symmetries enough to provide quantization according to the formulas by Li et al in Ref 10?

Similar to a local four-fold symmetry, the reflection symmetries are insufficient to provide quantization in a system without fine-tuning, i.e. in presence of a finite electron-lattice couplings, or more generally mode-lattice couplings.

I also have the following minor comments:

​2. Is Eq (7) the definition of $N_0$? If so, I don't see from the equation why $N_0$ would be an integer. On the other hand, if $N_0$ is defined from Ref 14, then why would it be quantized in some region defined by the length scale R? I might expect it to exponentially decay with R.

We use Eq. (7) to establish the asymptotic behavior of the defect charge in the thermodynamic limit $L \rightarrow \infty$ when sublattice symmetry is preserved. Eq. (7) is derived from Eqs. (5-6) and it is therefore not a definition, but $N_0 = \lim_{R \rightarrow \infty} \sum_{i,j=-R/2}^{R/2} \sum_{E_l=0} \sum_{\alpha=1}^4 \lvert \Psi_{\alpha l}^{ij} \rvert^2$ is.

We agree with the referee that for the charge to be quantized, the charge density must decay sufficiently quickly with $R$. We show in Figures 4(a-d) that the exponential decay of the defect charge is a consequence of sublattice symmetry.

​3. I don't understand the perturbative argument (spanning p4-5) explaining the quantization provided by sublattice symmetry: is the perturbation gluing the lattice together across the quadrant that was removed? If so, does the gluing consists of "normal" strength bonds, or small strength -- if the bonds are full strength, then is this really a small perturbation?

Yes, the perturbative argument in pages 4 and 5 considers gluing the lattice across the quadrant that was removed. Because anywhere away from the dislocation the lattice has sublattice symmetry, the only perturbation that remains is the dislocation center itself. While it is a strong perturbation, its effect is local because of the bulk gap.

​5. The authors reference the Riemann rearrangement theorem (top of p6) to argue that the defect charge may assume an arbitrary value. But I am not convinced of this argument: from what I can see, the Riemann rearrangement theorem says that the sum for q_{tot} in Eq (7) can be rearranged to reach an arbitrary value if it is arrange in any order. But physically, the unit cells closest to the defect should be included in the sum before terms further away. With this constraint, can the sum still reach an arbitrary value?

If a specific summation order is chosen, then the defect charge may reach an arbitrary finite or infinite value depending on the microscopic Hamitlonian details. This is shown in our results, where we always include the unit cells closest to the defect when computing the defect charge and obtain a non-quantized defect charge that depends on the Hamiltonian parameters. We are, however, unaware of the physical meaning of this quantity, nor which physical considerations dictate to perform integration in a specific order.

​6. I am confused by the caption to Fig 4(a), which is said to demonstrate the "lack of absolute convergence" for q_{tot}. But Fig 4(a) looks like it converges to a flat line. Perhaps the authors mean a lack of quantization?

Figure 4(a) shows the apparent convergence of the total charge. However, the value to which the charge appears to converge depends on the integrated area. This is a characteristic feature of conditional convergence without absolute convergence. Moreover, Figure 4(b) shows the lack of absolute convergence of the charge deviation. Figures (a-b) demonstrate that the defect charge is not quantized.

---

## Round 3 · Author Response

Resubmission letter for BOTP defects

We thank the referees for their feedback and for their overall positive evaluation. Below we list the detailed response to the referee inquiries, followed by the list of changes in the manuscript.

Response to referee 1

  1. For Fig 2(b), what is the corresponding filling for the charge density? In that case, zero energy states are also localized at the edge and it's unclear if the summation on the right-hand side of eq (7) is quantized.

We thank the referee for pointing out this omission. We compute the charge density at half-filling, and we have now clarified this in the caption of Fig. 2(b). Specifically, throughout the manuscript we use the Fermi energy $E_F=-0.5$, which lies in the bulk gap. We use the Eq. (7) to establish the properties of the defect in the thermodynamic limit $L \to \infty$. For that reason the sum in the Eq. (7) does not include the edge state contributions and yields the total number of defect zero modes. In the numerics we use finite systems, however, the gapless edge modes are exponentially localized on the edge, therefore the contribution inside the box centered on the defect far from the edges is negligible.

  1. Is there a good physics understanding for that convergence? e.g. is it because the profile of Wannier orbitals loses the symmetry when C4 is broken and portions of Wannier orbitals that fall into the integral region are therefore slightly different from the 1/2? Would the deviation depend on how large the C4 symmetry is broken?

According to the Riemann rearrangement theorem, the lack of absolute convergence means that the value of the total charge depends on the summation order of the local charge densities, and can be made to diverge or converge to an arbitrary value. In other words, in the absence of sublattice symmetry, the computed defect charge depends on the shape of the integrated area around the defect, and it is therefore undefined. We have now clarified this in text following Eq. (8). The magnitude of the local charge density indeed scales with the Hamiltonian gradients, as we describe in the paragraph containing Eq. (9).

Response to referee 2

  1. Could the author provide evidence that it is not a numerical error? Or, when the $\delta$ is larger, is the converged value far from 0.5?

The total defect charge deviates from $e/2$ due to the Riemann rearragement theorem, which guarantees that the specific defect charge value depend on the order of summation, and is therefore undefined. That the Riemann rearrangement theorem applies is supported by Figs. 4(b) and 4(d), as well as by the scaling argument in the paragraph containing Eq. 9. We have confirmed that with a fixed summation order defined by Eq. 4, the deviation of the apparent defect charge from the quantized value is proportional to the strength of sublattice symmetry breaking, as one would naturally expect.

  1. In Figures 2 (b-d), it seems that the charge is distributed on the boundary of systems. Do the systems have nontrivial edge states or edge charge?

To the best of our knowledge, the edge charge density in this system is not topologically protected.

  1. The authors said "We identify sublattice symmetry and not higher order topology as the origin of the previously reported charge quantization." Then, I wonder if the sublattice symmetry is crucial in the fractional disclination charge of rotation symmetric systems reported in Ref. [10]. Could the authors show the importance of sublattice symmetry through the models in Ref. [10]?

Our argument, when applied to disclinations, is as follows. Disclinations in purely 2D systems necessarily introduce strain, and therefore break $C_4$ symmetry due to electron-lattice coupling. A purely topological disclination that does not alter the local Hamiltonian and preserves local $C_4$ symmetry does guarantee charge quantization even without sublattice symmetry. However, this may only occur in a conical lattice, and not a two-dimensional one. Upon reviewing the manuscript we have realized that this point was not made in a sufficiently clear manner, and we have clarified the argument in the updated version. We have also confirmed that a topological disclination in the BBH model with $C_4$ symmetry and broken sublattice symmetry does preserve charge quantization.

  1. The authors should add the converged value of solid lines in Figure 3(a).

We have clarified in text following Eq. (8) that, according to the Riemann rearrangement theorem, the lack of absolute convergence means that the total charge converges to an arbitrary value depending on the shape of the integration region. Because the value is arbitrary, we think it is misleading to add the specific value to the figure.

  1. The authors should add plot legends in Figure 2.

We have added the legends and colorbar to the plots in the figure.

---

## Round 3 · List of Changes

• Emphasized that both topological defects necessarily break four-fold rotation symmetry even away from the defects in the abstract and throughout the text.
  • Added Figure 2 to show how the Wannier representation of the parametric defect indicates a $e/2$ defect charge.
  • Swapped the disclination and parametric defect paragraphs for a more streamlined presentation.
  • Added labels and colormap to Figure 3, together with clarifying that the systems are at half-filling.
  • Complemented the explanation on lack of quantization with the Riemann rearrangement theorem.
  • Clarified that if local four-fold rotation is present, the disclination charge is quantized regardless of sublattice symmetry.

We also attach a PDF file with the changes highlighted: https://surfdrive.surf.nl/files/index.php/s/YRgayz0uJQqJSKO

---

## Editorial Decision

published